# Circulated TGF-β1 and VEGF-A as Biomarkers for Fabry Disease-Associated Cardiomyopathy

**DOI:** 10.3390/cells12162102

**Published:** 2023-08-19

**Authors:** Margarita M. Ivanova, Julia Dao, Omar Abu Slayeh, Andrew Friedman, Ozlem Goker-Alpan

**Affiliations:** Lysosomal & Rare Disorders Research and Treatment Center, 3702 Pender Drive, Ste 170, Fairfax, VA 22030, USA

**Keywords:** Fabry disease, cardio, biomarkers, fibrosis, TGF-β1, VEGF, FGF2

## Abstract

Fabry disease (FD) is a lysosomal disorder caused by α-galactosidase A deficiency, resulting in the accumulation of globotriaosylceramide (Gb-3) and its metabolite globotriaosylsphingosine (Lyso-Gb-3). Cardiovascular complications and hypertrophic cardiomyopathy (HCM) are the most frequent manifestations of FD. While an echocardiogram and cardiac MRI are clinical tools to assess cardiac involvement, hypertrophic pattern variations and fibrosis make it crucial to identify biomarkers to predict early cardiac outcomes. This study aims to investigate potential biomarkers associated with HCM in FD: transforming growth factor-β1 (TGF-β1), TGF-β active form (a-TGF-β), vascular endothelial growth factor (VEGF-A), and fibroblast growth factor (FGF2) in 45 patients with FD, categorized into cohorts based on the HCM severity. TGF-β1, a-TGF-β, FGF2, and VEGF-A were elevated in FD. While the association of TGF-β1 with HCM was not gender-related, VEGF was elevated in males with FD and HCM. Female patients with abnormal electrocardiograms but without overt HCM also have elevated TGF-β1. Lyso-Gb3 is correlated with TGF-β1, VEGF-A, and a-TGF-β1. Elevation of TGF-β1 provides evidence of the chronic inflammatory state as a cause of myocardial fibrosis in FD patients; thus, it is a potential marker of early cardiac fibrosis detected even prior to hypertrophy. TGF-β1 and VEGF biomarkers may be prognostic indicators of adverse cardiovascular events in FD.

## 1. Introduction

Fabry disease (FD) is an X-linked lysosomal disorder wherein pathogenic variants of the *GLA* gene result in the deficiency of the enzyme α-galactosidase A (α–Gal A) (EC entry 3.2.1.22). The presentation of FD is quite heterogeneous and includes signs and symptoms related to cardiovascular disease, renal failure, and cerebrovascular complications, including ischemic or hemorrhagic strokes, all of which are associated with reduced quality of life and early mortality [1,2]. While FD occurs in a spectrum, there is a subgroup of patients with predominant cardiac involvement with a higher residual α–Gal A activity, a phenotype occasionally referred to as the “cardiac variant” of FD. Nevertheless, cardiovascular complications contribute substantially to morbidity and are the leading cause of premature death in both male and female patients with FD. Cardiovascular involvement in FD includes left ventricular hypertrophy (LVH), valvular disease, diastolic dysfunction, and microvascular angina [3]. However, hypertrophic cardiomyopathy (HCM) is the most common cardiac pathology and cause of death in patients with FD. 

α–Gal A catalyzes the lysosomal hydrolysis of globotriaosylceramide (Gb-3) to lactosylceramide and galactosylceramide (Gal-Gal-Cer) to galactosylceramide (Gal-Cer) [4,5]. The deficiency of α–Gal A leads to an accumulation of Gb-3 and its metabolites, globotriaosylsphingosine (Lyso-Gb-3) and Gal-Gal-Cer, in the lysosomes [6]. The deposition of Gb-3 and Lyso-Gb-3 within the myocardium was suggested to affect cardiac function with consequential progressive cardiovascular involvement [7]. Gb-3 accumulation was shown in the cardiac valves, cardiomyocytes, nerves, and coronary arteries. Gb-3 and Lyso-Gb-3 accumulation in the cardiac tissue increases the release of inflammatory molecules, transforming growth factors, and triggers a cascade of events leading to inflammation and end-stage fibrosis [8,9]. 

The superfamily of transforming growth factor β (TGF-β) is a profibrotic cytokine/growth factor, where TGF-β1 is the isoform of TGF-β in cardiac tissue [10]. It is well established that the elevation of TGF-β1 is involved in cardiac fibrosis, cardiomyocyte apoptosis, and cardiac hypertrophy. In hypertensive patients, serum TGF-β1 levels were shown to correlate with left ventricular hypertrophy [11]; whereas in patients with aortic stenosis, increased levels of TGF-β1 were associated with both higher transvalvular gradients and hypertrophy [12]. Most cells in the myocardium release TGF-β1; however, a significant source of secreted TGF-β1 is derived from the macrophages that infiltrate the myocardium to engulf the damaged cardiomyocytes after cardiomyocyte apoptosis [13]. The precursor of TGF-β1 has three functional regions: the N-terminal 29 amino acid peptide required for secretion from a cell, a 249 amino acid pro-region (latency-associated peptide or LAP), and a 112 amino acid C-terminal region that becomes the active-TGF-β1 [14,15]. Therefore, plasma contains a high concentration of latent TGF-β1 and a low concentration of active-TGF-β1. The active-TGF-β1 binds to the TGF-β receptor and exerts biological functions [15].

With cardiac hypertrophy, there is also an expansion of the coronary vasculature to maintain a sufficient supply of oxygen and nutrients. Thus, activation of coronary angiogenesis plays a vital role in the pathological cascade leading to cardiac hypertrophy. Vascular endothelial growth factor (VEGF-A) and fibroblast growth factor (FGF-2) are the most potent activators of angiogenesis, stimulating the proliferation and migration of endothelial cells to generate new blood vessels [16]. Proteomic analysis of plasma from patients with FD showed an elevation of angiogenesis-related proteins, VEGF-A, VEGF-C, and FGF2 [17]. 

This study aimed to assess the plasma levels of TGF-β1, active-TGF-β1, VEGF-A, and FGF2 in patients with FD and their correlation with the other progression of FD-associated cardiomyopathy. Moreover, the FD-specific biomarker, Lyso-Gb3, was evaluated along with fibrosis/angiogenesis markers to demonstrate the role of circulating sphingolipids with the presence and severity of cardiac involvement in patients with FD. 

## 2. Materials and Methods

### 2.1. Subjects

In an IRB-approved protocol NCT04724083, 45 patients diagnosed with FD (21 males and 24 females) (mean age: 40 ± 14 years) and 20 healthy controls (10 males and 10 females) (mean age: 48 ± 11 years) were studied. Study participants were categorized into two groups: FD subjects with cardiomyopathy (n = 28, mean age 44 ± 18), and FD subjects without cardiomyopathy (n = 17, mean age 35.1 ± 13). The “borderline” group was comprised of patients with an abnormal EKG (n = 6) (three females and three males). The mean age of patients diagnosed with HCM was 34.3 ± 13.5 years for males and 44.4 ± 10.9 for females. All male patients with FD received Enzyme Replacement Therapy (ERT), while only 9 female patients with HCM were on ERT. Three female subjects received chaperone therapy, one switched from ERT to chaperone therapy, and two were without treatment (naïve). Measurements of plasma and urine lyso-Gb3, clinical biomarkers of FD, were obtained from the patient’s medical histories. 

The demographic data, including age, sex, genotypes, FD clinical biomarkers, and treatment status, are presented in Table 1. Clinical characteristics of cardio parameters were stratified into four groups (Table 2). 

### 2.2. Enzyme-Linked Immunosorbent Assay (ELISA):

Venous blood samples were collected in EDTA vacutainer tubes, and plasma levels of growth factors were measured using commercially available ELISA kits. The FGF2 concentration was measured in 50 µL of plasma using FGF2 ELISA. VEGF-A was measured in 50 µL of plasma using a human VEGF-A ELISA kit. TGF-β1 was measured in 0.5 µL of plasma using a human TGF-β1 ELISA kit (Origene, Rockville, MD, USA). Free Active TGF-β1was measured in plasma using a human LEGEND MAX™ Free Active TGF-β1 ELISA Kit (Biolegend, San Diego, CA, USA).

### 2.3. Statistical Analysis

Statistical analysis was performed using Graph Prizm (GraphPad, San Diego, CA, USA). Differences between the two groups were tested via Student’s *t*-test (unpaired) or F-test. The groups were compared using a one-way analysis of variance (ANOVA) followed by the Kruskal–Wallis test. The relationships between TGFβ1, active-TGFβ, VEGF-A, FGF2, and clinical biomarkers (plasma and urine Lyso-Gb3) were determined using the Pearson or Spearman correlation technique. 

## 3. Results

### 3.1. Circulating Levels of TGF-β1 and the Active Form of TGF-β1 Are Elevated in Patients with Fabry Disease

Transforming growth factor β1 (TGF-β1), a proinflammatory cytokine, contributes to cardiac fibrosis by switching cardiac fibroblasts to myofibroblasts that are responsible for collagen deposition within the myocardium [10,18]. Thus, to examine whether TGF-β1 correlates with cardiomyopathy in FD, the plasma level of TGF-β1 and its active form were measured using commercial ELISAs. Analysis of healthy control plasma and plasma from FD cohorts demonstrated that the level of TGF-β1 was significantly higher in FD (Figure 1A, Table 3). Furthermore, the finding that TGF-β1 is elevated in FD patients remained unchanged after stratifying by gender (Figure 1B).

The precursor of TGF-β1 contains 390 amino acids with an N-terminal signal peptide of 29 amino acids, which is required for cellular secretion. A 249 amino acid pro-region (latency-associated peptide or LAP) and a 112 amino acid C-terminal region become the active TGF-β1 upon activation [19]. The free active form of TGF-β1 binds to the TGF-β1 receptor and exerts biological functions. Analysis of healthy control plasma and plasma from FD cohorts demonstrated that the plasma level of free active TGF-β1 was higher in FD patients than in healthy controls (Figure 1C). However, increased levels of the active form of TGF-β1 were observed in females but not in males (Figure 1D and Table 3). In addition, we noticed a wide distribution of the levels of the active TGF-β1 in healthy male controls, resulting in no significant elevation of active TGF-β1 in FD males. 

Next, we investigated the correlation between TGF-β1 and the active form of TGF-β1. A positive linear correlation was observed between plasma levels of TGF-β1 and the active form of TGF-β1 in healthy controls but not in FD patients (Figure 1E,F).

### 3.2. Circulating Angiogenic Biomarkers FGF2 and VEGF-A Are Elevated in FD

Fibroblast growth factor 2 (FGF2) is a growth factor that functions in angiogenesis, wound healing, tissue repair, and heart, bone, and brain morphogenesis. Moreover, FGF2 is upregulated in response to inflammatory stimuli and promotes the differentiation of fibroblasts and cardiomyocytes [20,21]. Evaluation of healthy control plasma and plasma from FD patients demonstrated that the levels of FGF2 were significantly higher in FD (Figure 2A and Table 3). A more detailed analysis verified that FGF2 levels were elevated in females with FD but not in males (Figure 2B). 

Vascular endothelial growth factor (VEGF-A) is an angiogenic factor originally described as an essential growth factor for vascular endothelial cells. In the general population, increased VEGF-A is associated with structural and functional parameters in patients with HCM. Elevation of VEGF-A in FD has been described and suggested as a response to vascular damage [22]. In this study, VEGF-A was significantly higher in the FD cohort than in the healthy controls (Figure 2C and Table 3). Further analysis demonstrated that male patients with FD showed a significant elevation of plasma VEGF-A compared with healthy controls (Figure 2D). Furthermore, VEGF-A levels in female patients with FD demonstrated a less prominent difference when compared to healthy female controls (F-test significant differences). A positive linear correlation was observed between plasma levels of VEGF-A and FGF2 in healthy controls (Figure 2E). However, we did not observe the same correlation in FD patients (Figure 2E,F). Additionally, age-related analyses of growth factors showed that FGF2 and VEGF-A decreased in healthy males but not in healthy females (Figure 2G–I,K). However, comparing FD males with healthy controls showed an absent correlation between age and FGF2 or VEGF-A levels in FD (Figure 2L,M).

### 3.3. The Association between Growth Factors and Lyso-Gb-3 Accumulation in FD

Gb-3, the substrate of α-Gal A that could be detected both in urine and plasma, is mostly suitable for identifying males with classical FD and HCM cardiac rhythm disturbances, progressive renal failure, and/or strokes [20]. Plasma lyso-Gb-3, the deacylated derivative of Gb-3, has been designated as a hallmark of FD and could be useful as an FD-specific biomarker, particularly in females with normal or borderline α-Gal A activity [21,22]. Furthermore, the deposition of Lyso-Gb-3 within the myocardium has been shown to affect cardiac functions and correlates with cardiovascular pathology [7,23]. Thus, we investigated whether plasma- and urine-Lyso-Gb-3 levels correlated with the circulating growth factors. 

First, we analyzed the correlation between plasma Lyso-Gb3 and urine Lyso-Gb-3 in all FD patients. A positive linear correlation was observed between plasma and urine Lyso-Gb-3 (Figure 3A). FD patients with HCM demonstrated an elevated Lyso-Gb-3 in plasma and urine (Figure 3B,C). Further analysis revealed that plasma Lyso-Gb-3 was significantly higher in males but not females with FD, especially in males with mild and moderate/severe stages of HCM (Figure 3D). Regarding urine Lyso-Gb3, the majority of FD males with HCM have higher levels of urine Lyso-Gb-3; however, because one patient without HCM had an elevated level of urine Lyos-Gb-3, no statistical differences were observed (Figure 3E). Among male patients with HCM, 7 out of the 11 exhibited the classic FD form, and 3 were excluded from the analysis because of a kidney transplant. However, we did not observe any differences in Lyso-Gb-3 levels between the classic and non-classic FD or mild and moderate/severe stages of HMC. Noteworthy, all of these patients were on long-term therapy, and it is known that ERT reduces Lyso-Gb-3 levels in classic Fabry males and stabilizes in most of the late-onset FD and FD females [23]. 

Next, we analyzed the correlation between the growth factors (TGF-β1, active-TGFβ, FGF2, and VEGF-A) and plasma/urine Lyso-Gb-3 levels. Pearson analysis demonstrated a positive linear correlation between plasma Lyso-Gb-3 and TGF-β1 as well as plasma Lyso-Gb-3 and VEGF-A (Table 4, Figure 4). Moreover, Spearman correlation analysis supported the monotonic correlation between plasma-Lyso-Gb-3 and TGF-β1. Surprisingly, the active TGFβ1 demonstrated the linear and monotonic correlation with urine Lyso-Gb-3, but not with plasma Lyso-Gb-3. Plasma FGF2 did not correlate with plasma or urine Lyso-Gb-3 levels (Table 4, Figure 4).

### 3.4. Plasma TGF-β1 Is Elevated in FD Patients with Abnormal EKG and HCM

To further assess the correlation between plasma TGF-β1 and HCM, the FD cohort was categorized into groups according to the HCM status based on echocardiogram and cardiac MRI results (Table 1). The cohort was divided into two groups: patients without HCM (No HCM), including the subgroup “borderline,” and the second group, patients with mild, moderate, and severe forms of HCM (HCM+). Next, the group “No HCM” was split into subgroups: patients with normal cardiac parameters and patients with abnormal EKG only. The abnormal EKG description included sinus bradycardia with left atrial enlargement, non-specific T wave abnormalities, and first-degree heart block. 

Statistical analysis demonstrated a significantly elevated level of TGF-β1 in FD cohorts “No HCM” and “HCM” as compared to healthy controls (Figure 5A). The same results were confirmed in gender-divided cohorts. FD females and males have significantly elevated levels of TGF-β1 compared with healthy controls (Figure 5B,C). When comparing the FD cohorts with HCM and without HCM, TGF-β1 showed no significant differences. However, a significantly high level of TGF-β1 was observed between the borderline group and healthy controls, and the borderline group and “no HCM” FD cohorts (Figure 5D). 

Further analysis demonstrated that male patients with FD displayed an elevated level of TGF-β1 in both cohorts “No HCM-normal EKG” and with HCM (Figure 5E); in females, TGF-β1 was significantly elevated in the cohorts where there is evidence of cardiac involvement, including “No HCM-abnormal EKG” and with HCM (Figure 5F). Thus, plasma TGF-β1 as a biomarker is a promising candidate for early detection of HCM, especially in FD women. However, we would like to point out that analysis should be interpreted cautiously due to the relatively small number of subjects in the subgroups “no HCM” and “no HCM and abnormal EKG”.

### 3.5. Active TGF-β Does Not Correlate with Cardiomyopathy in Fabry Disease

Statistical analysis (ANOVA, *t*-test, or F-test) showed no differences in active-TGFβ levels between FD patients with and without HCM. Gender-stratified analysis similarly did not demonstrate any differences for TGFβ among the different cohorts (Figure 6). 

### 3.6. The Association of VEGF-A and FGF2 with FD Cardiomyopathy

Next, to study the effects of VEGF-A on cardiomyopathy in FD, the levels of VEGF-A in FD patients with and without HCM were compared (Figure 7). If marked that the reference of the highest normal VEGF-A is 100 pg/mL, approximately 42% of FD patients with and without HCM have elevated VEGF-A. Moreover, the Student *t*-test showed a significant difference between the healthy controls and FD cohorts with HCM. Analysis of variance F-test showed a significant difference between “no HCM” and “HCM+” in FD (Figure 7A). 

Further analysis stratified by sex showed a significant difference between males and females: 50% of FD females without HCM have an elevated VEGF-A; while only 28% of FD males without HCM demonstrated elevated VEGF-A. However, in FD patients with HCM, we observed the opposite results: only 28% of FD females with HCM have elevated VEGF-A; while more than half of FD males (57%) with HCM have increased VEGF-A levels. Additionally, FD male patients with HCM demonstrated a significantly elevated VEGF-A compared to healthy controls (Figure 7B). However, in females, patients without HCM showed a considerably elevated VEGF-A compared to healthy controls (Figure 7C). Another marker of angiogenesis, FGF2, did not correlate with HCM in Fabry disease (Figure 7D). Surprisingly, statistical analysis showed a significantly increased level of FGF2 in FD females without HCM. The age of female patients with the highest level of FGF2 was varied (18 to 68 years old), and only one patient has proteinuria. Thus, the elevation of FGF2 is not associated with age or kidney dysfunction. 

## 4. Discussion

Hypertrophic cardiomyopathy is one of the most common causes of morbidity and one of the main causes of mortality in FD, with cardiac-related death occurring at a mean age of 55 years in men and at 66 years in women [24,25,26]. Overall, 60% of male and 50% of female patients with FD have cardiac symptoms, with a mean age of onset 29.2 ± 4.4 and 34.5 ± 17.6 years, respectively [27]. In our cohort, the mean age of patients diagnosed with HCM was 34.3 ± 13.5 years for males and 44.4 ± 10.9 for females. The development of HCM at later ages could be due to the fact that patients were on long-term FD-specific therapy. Still, some patients who received ERT were diagnosed with HCM during young adulthood (<25 years). 

Cardiovascular magnetic resonance (CMR) and echocardiography are considered the standard imaging modalities to identify and monitor patients with FD and cardiac involvement. Certain cardio measurements are suggested to be related to GLA deficiency and the accumulation of the substrate, Gb-3; for example, Gb-3 accumulation in the cells has been proposed to shorten myocardial T1 duration with unique features on T1 mapping CMR sequences [25,28]. The advantages of imaging for diagnostic cardiomyopathy include the visualization of the LV wall and thickness or using late gadolinium enhancement (LGE) to identify fibrosis. However, sometimes, interpretation of functional analysis or LGE in patients with HCM can be challenging. Furthermore, age, gender, or racial/ethnic differences may contribute to incorrect interpretation of the imaging data. Thus, differences in the hypertrophy morphology, magnitude of fibrosis, and limitations of CMR and echocardiogram imaging highlight the importance of finding other biomarkers related to primary disease pathology which could be assessed easily and at a low cost, i.e., not requiring advanced technical abilities. Such biomarkers could inform the clinician by predicting patient outcomes and impacting the management of patients with FD.

The involvement of the myocardium in FD is suggested to start early and is characterized by progressive lowering of transverse relaxation time 1 (T1) on cardiac MRI (CMR) even without LVH. Chronic inflammation and hypertrophy are associated with low T1 values and the initiation of LVH in males; in T2 mapping, there is evidence of inflammation with Late Gadolinium Enhancement (LGE), particularly observed in females as an early event [3,26]. As a final point, progressive inflammation and hypertrophy activate interstitial and perivascular or replacement fibrosis [29,30]. The mechanism of activation and the development of cardiac fibrosis in FD is still not thoroughly investigated. Generally, myocardial fibrosis is divided into replacement fibrosis and interstitial and perivascular fibrosis. “Replacement fibrosis” refers to when collagen-based scars replace necrotic cardiomyocytes. “Interstitial and perivascular fibrosis” is the expansion of the endomysial and perimysial space without significant cardiomyocyte loss [31]. Male patients with classical FD develop replacement fibrosis, usually in the area of posterolateral segments of the basal myocardium [3,26]. Female patients could start to develop interstitial and perivascular fibrosis without detectable early signs of cardiac involvement. Thus, the early and precise diagnosis of FD cardiomyopathy is critical to prevent irreversible end-organ damage associated with fibrosis.

The end result of the pathological process in HCM is myocardial fibrosis. The cellular biology of fibrosis involves signaling and activation of cardiac fibroblasts, which is responsible for the deposition of collagens in the overloaded myocardium. Cardiac fibroblasts comprise around 20% of non-myocyte cells in the heart, and in the inactive stage, fibroblasts have no significant changes in cell numbers and proliferative activities [32]. In response to stress or injury, the TGF-β/Smad3 signaling pathway activates cardiac myofibroblasts to maintain the structural integrity of the myocardium by contracting surrounding tissue and thus forming the fibrotic tissue. TGF-β1 induces the differentiation of cardiac fibroblasts to myofibroblasts, increasing the production of extracellular matrix, fibronectin, and collagen in the heart tissue [33,34]. In this case, the source of circulating TGF-β1 can be the fibroblasts themselves, as well as macrophages or other immune cells infiltrating the heart tissue. Elevated plasma TGF-β1 is associated with an increased risk of heart failure and myocardial remodeling in the general population [35,36]. The main finding in this study is that TGF-β1 is significantly increased in patients with FD and is associated with HCM, even at the early stages. Interestingly, female patients with FD and normal LVPWD and LV mass but with EKG changes, including bradycardia or T wave abnormalities, have significantly elevated TGF-β1 levels. Thus, TGF-β1 can be a promising early biomarker for FD-associated HCM even before the structural alterations of the heart tissue could be detected through the abovementioned imaging modalities. 

Cardiac hypertrophy corresponds to the expansion of the coronary vasculature to maintain a sufficient supply of oxygen and nutrients. Therefore, activation of coronary angiogenesis is a necessary step in cardiac vascularization and hypertrophy. VEGF-A and FGF-2 are the activators of angiogenesis that stimulate the proliferation and migration of endothelial cells. Previously, the proteomics-based analysis showed an elevation of VEGF-A and FGF2 in FD patients compared to healthy controls [17]. It was hypothesized that the elevation of VEGF-A and FGF2 might have further contributed to vascular damage [37]. We noted that plasma FGF2 levels differed between genders and were particularly increased in female patients with FD. Further analysis showed that more than half of FD males (57%) with HCM, not females, have increased VEGF-A levels; in females, on the contrary, about 50% without HCM have elevated VEGF-A. The multifunctional role of VEGF-A could be a reason for the lack of a robust correlation with a cardiac finding. VEGF-A is expressed by many cells, including endothelial cells and cardiac fibroblasts, in the presence of immune activation secreted by macrophages, lymphocytes, and monocytes, and many factors that regulate this expression. Moreover, plasma VEGF-A is associated with body mass index, blood pressure, inflammation, etc. In the kidney, VEGF-A is produced by podocytes and is associated with chronic kidney diseases [38,39]. Therefore, the elevation of VEGF-A in FD may result in the development of cardiac fibrosis, kidney fibrosis or inflammation, or a combination of all pathological events. 

The accumulation of Gb-3 in FD within the cardiomyocytes impairs energy metabolism, altering the electrophysiology, impacting the valves’ structure, and leading to LV hypertrophy [7,26]. Gb-3 is deacetylated into water-soluble Lyso-Gb-3 in the lysosome, which easily diffuses through the membranes and can be measured in circulation and other various body fluids. Multiple in vitro studies correlate the deposition of Gb-3 and Lyso-Gb-3 in cardiomyocytes with profibrotic behavior that leads to myocardial fibrosis [40,41]. Moreover, levels of Lyso-Gb-3 are associated with important clinical events, including a positive correlation with LVH and white matter changes [22,42,43]. Our results show that most FD male patients, especially with mild or moderate/severe forms of HCM, have significantly elevated levels of Lyso-Gb-3. Furthermore, plasma Lyso-Gb-3 has a positive linear correlation with TGF-β1, and urine Lyso-Gb-3 has a positive linear correlation with the active-TGF-β1 in FD patients. This result is in line with Fabry mouse models demonstrating that TGF-β1 plays a primary role in organ damage, such as in FD nephropathy. Similarly, in vitro, exposure to Lyso-Gb-3 increases the expression of TGF-β1 in renal cell lines and FD podocytes [8,44,45,46]. 

A recent study with proteomic analysis showed that plasma VEGF-A and FGF2 are elevated in patients with FD, but a positive correlation between VEGF-A—though not FGF2—and plasma Lyso-Gb-3 was observed in the samples from patients with classic FD who were naïve [17]. We confirmed that VEGF-A, similar to TGF-β1, correlated with plasma Lyso-Gb-3 regardless of the treatment status. The absence of correlation between FGF2 and Lyso-Gb-3 levels highlights the differences in the regulation mechanisms of VEGF-A and FGF2. Moreover, age-related and gender-related differences in VEGF-A and FGF2 secretion exhibit the potential complexity of utilizing these molecules as biomarkers in FD [47,48]. However, at the same time, VEGF-A and FGF2 levels could help in understanding the mechanisms for the induction of fibrogenic signals in FD. 

The association analysis among the observed changes in biomarkers, clinical parameters, and sex showed noteworthy patterns. Female patients with FD present with a wide range of clinical symptoms of variable disease onset and severity. Gender differences in the mechanism of activation of cardiac fibrosis could be due to different factors. One factor reflects the residual α-gal A activity in females, and the second is a sex-specific difference in cardiac function and the aging of the heart [29,49]. Thus, in the general population, myocardial fibrosis is more common in the aging male heart and is often delayed in women; moreover, the mechanism of fibrosis is distinct between males and females [50]. For example, in males, it has been suggested that regional fibrosis occurs through apoptosis and is concomitant with reactive interstitial and perivascular fibrosis [49,51]. Women with dilated cardiomyopathy have a lower rate of cardiac fibrosis, apoptosis, and necrosis [50,52]. In FD, male patients first develop LV hypertrophy and then fibrosis. In females, however, fibrosis could occur without hypertrophy. It has been shown that about 25% of female patients with FD exhibit enhancement on CMR without an increase in LV mass. On the contrary, in male patients, late enhancement is always associated with the presence of LVH [53]. Thus, the increased levels of TGF-β1 in FD females without overt HCM may explain the occult initiation of fibrosis. FD females, not males, have elevated levels of TGF-β1 and an active-TGF-β. The active-TGF-β is a pro-migratory factor that recruits myofibroblasts for tissue repair/remodeling in response to inflammation or injury [31,54]. Clinical assessments utilizing Lyso-Gb-3 levels in female patients with FD are also problematic. Lyso-Gb-3 levels in the peripheral circulation is proposed to be derived from multiple organs, and due to the mostly nonuniform nature of the organ involvement in female patients, there is often a lack of correlation in the clinical events with Lyso-Gb-3 levels [23,43,55,56]. This study also demonstrated that Lyso-Gb-3 levels do not correlate with LVH or HCM in female patients with FD. Thus, new surrogate diagnostic markers are required to monitor cardiac involvement in females in combination with MRI. 

## 5. Conclusions

The elevation of TGF-β1 and active-TGF-β1 associated with Lyso-Gb-3 elevation provides evidence of a chronic inflammatory state and the activation of fibrosis in FD patients. Angiogenesis biomarker, VEGF, correlates with plasma Lyso-Gb-3 and is associated with hypertrophic cardiomyopathy in FD patients. Thus, serum TGF-β1 and VEGF are predictive biomarkers for adverse cardiovascular events in FD. Gender differences in the secretion of TGF-β1, VEGF, and FGF2 can explain patterns of cardiac involvement in male vs. female FD patients, with fibrosis occurring early in the course in females. 

## Figures and Tables

**Figure 1 cells-12-02102-f001:**
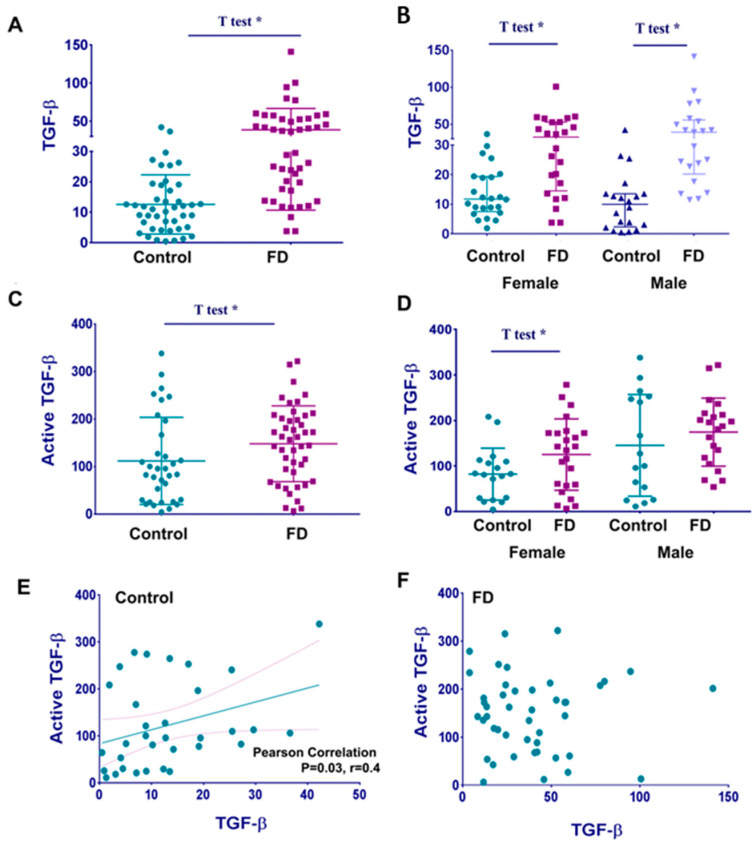
Circulating levels of TGF-β1 and “active form of TGF-β1”. (**A**) TGF-β1 levels in plasma from healthy controls and patients with FD are compared. Statistical analysis using an unpaired *t*-test and F-test to compare cohorts demonstrated a significant difference between control and FD cohorts. TGF-β1 (ng/mL) represented as a mean ± SDEV, *p* < 0.05 *t*-test. (**B**) The stratification of TGF-β1 based on gender and cohort groups showed that female and male patients with FD had a significantly higher level of TGF-β1 compared to female and male healthy controls. *p* < 0.05, *t*-test and F-test. (**C**) The active form of TGF-β1 levels in healthy control vs. patients with FD. Statistical analysis using an unpaired *t*-test and F-test to compare cohorts demonstrated no difference between the control and FD. (**D**) An active form of TGF β1 level in control females and female patients with FD, control males, and male patients with FD. (**E**) Scatterplot analysis of correlation of TGF-β1 and active TGF-β1 in plasma of healthy controls. * *p* < 0.03, Pearson correlation tests, two tails. (**F**) Scatterplot analysis of correlation of TGF-β1 and “active-TGF-β1” in plasma from patients with FD. No significant correlation between the two biomarkers.

**Figure 2 cells-12-02102-f002:**
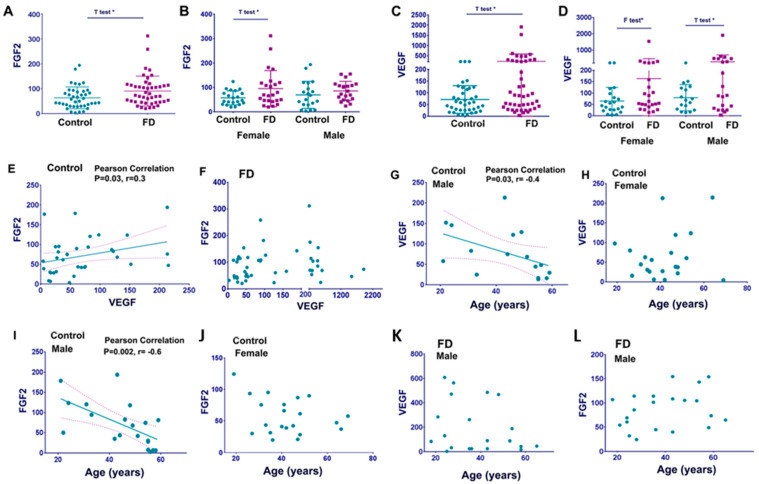
Circulated levels of FGF2 and VEGF-A. (**A**) FGF2 levels control vs. FD. Statistical analysis using an unpaired *t*-test demonstrated a significant difference between control and FD cohorts. *p* < 0.05 *t*-test. FGF2 pg/mL represented as a mean ± SDEV. (**B**) FGF2 levels in control females and female patients with FD, control males, and male patients with FD. *p* < 0.05 F-test, comparison control females vs. FD females. (**C**) VEGF-A levels control vs. FD. Statistical analysis using an unpaired *t*-test and F-test to compare cohorts demonstrated a significant difference between the healthy control and FD. VEGF-A pg/mL represented as a mean ± SDEV. (**D**) VEGF-A level in control vs. FD in gender-divided groups. *p* < 0.05 *t*-test, comparison control males vs. FD males, *p* < 0.05 F-test, comparison control females vs. FD females. (**E**,**F**) Scatterplot analysis of the correlation of FGF2 and VEGF in healthy controls (**E**) and FD (**F**). (**G**,**H**) Scatterplot analysis of correlation of age (years) and VEGF in healthy controls: males (**G**) and females (**H**). (**I**,**J**) Scatterplot analysis of correlation of age (years) and FGF2 in healthy controls: males (**I**) and females (**J**). (**K**,**L**) Scatterplot analysis of age vs. VEGF (**K**) and age vs. FGF2 (**L**) in male patients with FD showed an absent correlation between age and circulated VEGF and FGF2, respectively. * *p* < 0.05.

**Figure 3 cells-12-02102-f003:**
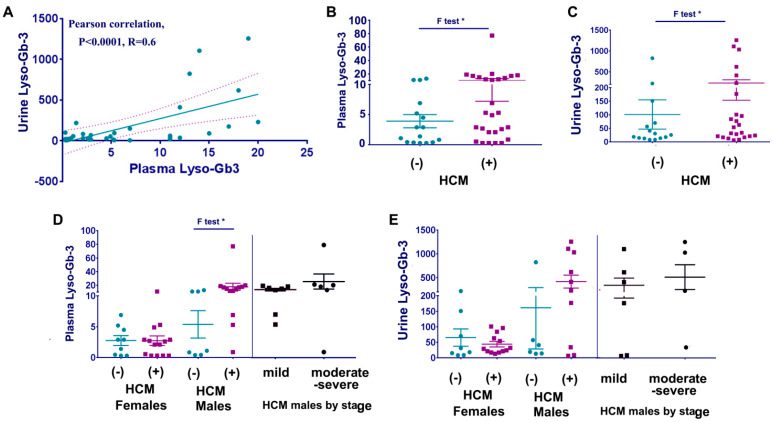
Relationship between plasma and urine lyso-Gb-3 and HCM in FD cohort. (**A**) Scatterplot analysis of correlation of Lyso-Gb-3 in plasma and urine. * *p* < 0.0001 Pearson tests, 90%, one tail. (**B**) Comparing plasma lyso-Gb-3 in FD patients with and without HCM. Plasma lyso-Gb-3 (ng/mL) represented as a mean ± SDEV, *p* < 0.05 F-test. All participants were under ERT. (**C**) Comparing urine lyso-Gb-3 in FD patients with and without HCM. Lyso-Gb-3 (µg/mmol) represented as a mean ± SDEV, *p* < 0.05 F-test. All participants were under ERT. (**D**) Comparing plasma lyso-Gb-3 in female and male FD patients with and without HCM. FD males with HCM stratified further to mild and moderate-severe stages of HCM. *p* < 0.05 F-test. (**E**) Comparing urine lyso-Gb-3 in female and male FD patients with and without HCM. FD males with HCM stratified further to mild and moderate-severe stages of HCM.

**Figure 4 cells-12-02102-f004:**
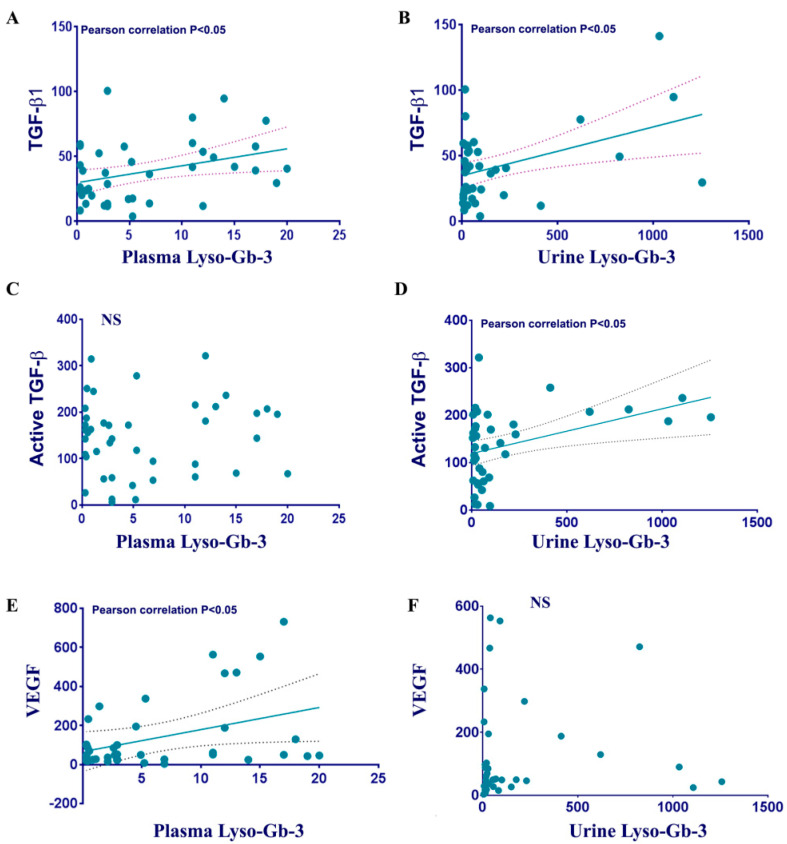
Correlation analysis between growth factor and accumulation of Lyso-Gb-3 in FD. (**A**,**B**) Scatterplot analysis of correlation of TGF-β1. Pearson correlation, two tail *p* < 0.05. (**C**,**D**) Scatterplot analysis of correlation of active TGFβ. Pearson correlation, two tail *p* < 0.05. (**E**,**F**) Correlation analysis between VEGF-A with plasma Lyso-Gb-3 (left panel) or urine Lyso-Gb-3 (right panel). Pearson correlation, one tail *p* < 0.05.

**Figure 5 cells-12-02102-f005:**
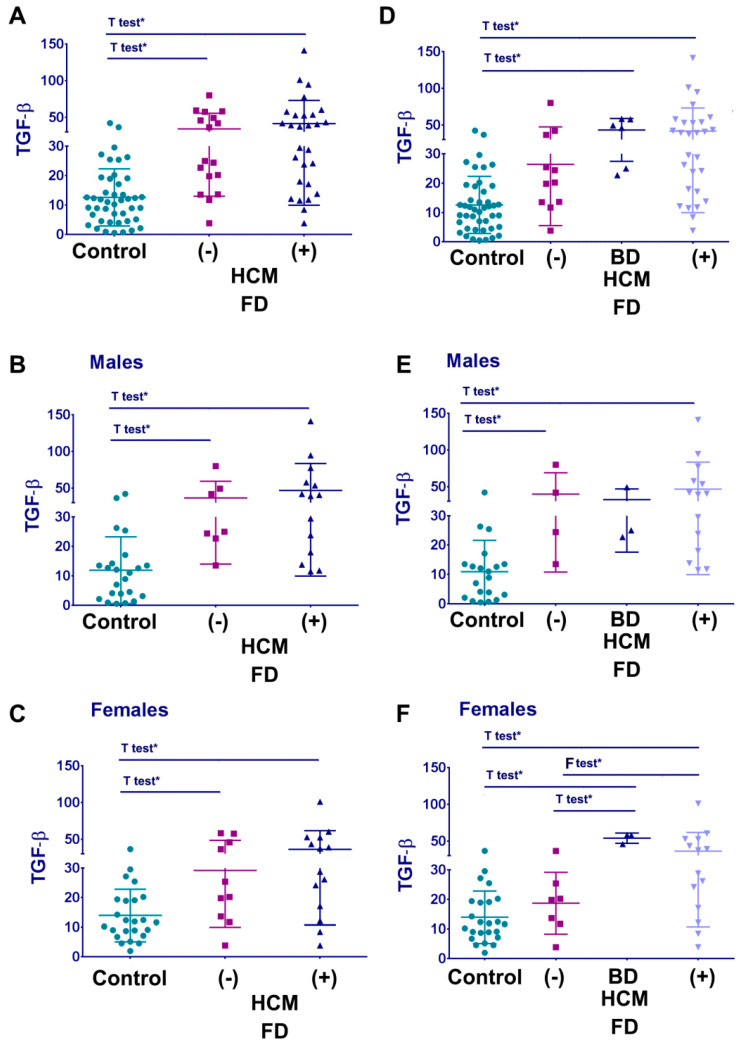
Plasma TGF-β1 and HCM in FD. (**A**–**C**) Level of TGF-β1 in the healthy control cohort (control), FD cohort without HCM, and with HCM. Graphs represent all FD patients (**A**), only males (**B**), and females (**C**). (**D**–**F**) Comparing levels of TGF-β1 in the healthy control, FD cohort without HCM and normal EKG, without HCM but abnormal EKG, and FD patients with HCM. Graphs represent all FD patients (**D**), males (**E**), and females (**F**). Data represented in pg/mL, mean ± SDEV, Kruskal–Wallis test. *p* < 0.0001 (significant difference for all graphs), * *t*-test or F-test: *p* < 0.05.

**Figure 6 cells-12-02102-f006:**
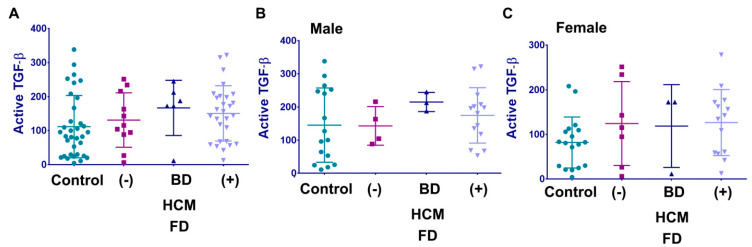
Active plasma Circulated active form of TGFβ and cardiomyopathy in patients with FD. (**A**) Active TGF-β level in the healthy control cohort control, FD cohort without HCM and normal EKG (−), without HCM but abnormal EKG (BD), and FD patients with HCM (+). Data represented as mean + STDEV. (**B**,**C**). The same cohorts for active TGF-β analysis are for male (**B**) and female (**C**) cohorts.

**Figure 7 cells-12-02102-f007:**
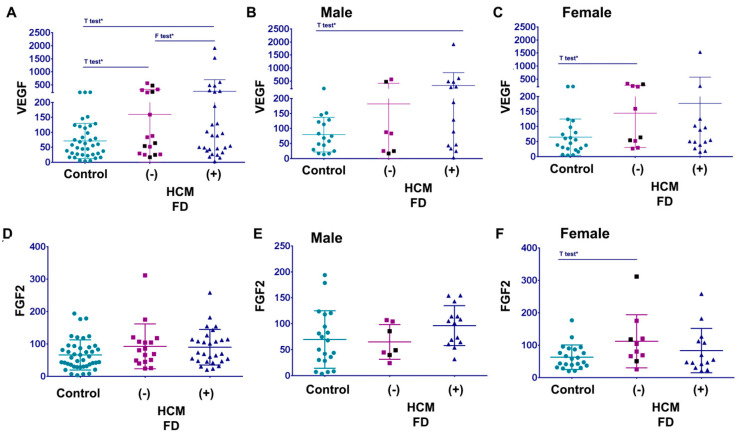
Angiogenic biomarkers VEGF-A and FGF2 in FD patients with cardiomyopathy: (**A**) VEGF-A level in the healthy control cohort (control), FD cohort without HCM, and FD patients with HCM. Purple square is the group: no HCM and normal EKG. The black square is the group: no HCM and abnormal EKG. (**B**,**C**) The same cohorts for VEGF-A analysis are for male (**B**) and female (**C**) groups. Data represented as mean + STDEV. One tail *t*-test *p* < 0.05. F test *p* < 0.05. (**D**) FGF2 level in the healthy control, FD cohort without HCM (HCM−) and with HCM (HCM+). Purple square is the group: no HCM and normal EKG. The black square is the group: no HCM and abnormal EKG. (**E**,**F**) FGF2 analyses are for male (**E**) and female (**F**) FD patients. Data are represented as mean + STDEV. One tail *t*-test * *p* < 0.05.

**Table 1 cells-12-02102-t001:** Demographics, genotypes, and other clinical and laboratory characteristics of patients with FD. ERT—enzyme replacement therapy; CT—chaperone therapy. * ERT: Repragal and later Fabrazyme.

No.	Age (Years)	Sex M/F	Genotype (Allele 1/Allee2)	Plasma Lyso-Gb-3 ng/mL	Urine Lyso-Gb3 µg/mmol/cr	Therapy (Years)
FD subjects without cardiomyopathy						
1	25	F	c.718_719del	0.46	8.3	* ERT > 10
2	45	F	C223Y	2.9	70	ERT 5–10
3	46	F	c.966DelC	-	-	ERT < 5
4	21	F	R49P	2.9	<6	ERT > 10
5	18	F	R363 H	0.3	8.6	ERT < 5; CT-2; ERT 2
6	27	F	c1033_1034delTC	1.4	220	* ERT > 10
7	33	F	c.1072_1074del	6.9	151	naïve
8	28	M	422C>T	11	41	ERT > 10
9	18	M	R49P	11	19	* ERT > 10
10	36	M	M296V	0.84	102	ERT 5
11	53	M	A143T	0.39	14	CT > 10
12	39	F	N215S	0.3	18	ERT-2; CT-4
13	56	F	G328R	5.2	15	* ERT > 10
14	68	F	C2233Y	4.5	31	ERT < 5
15	26	M	c.718_719delAA	13	825	ERT < 5
16	58	M	T41I	0.4	13	CT < 5
17	42	M	N215S	1.1	57	ERT > 10
FD subjects with hypertrophic cardiomyopathy (HCM)						
18	57	F	R49P	4.9	40	* ERT > 10
19	41	F	R301X	11	63	ERT < 5
20	46	F	c.718_719delAA	2.9	-	*ERT > 10
21	46	F	Q279E	2.1	32	CT < 5
22	44	F	Arg227Gln	2.9	17	ERT < 5
23	21	F	c1033_1034delTC	2.1	84	CT < 5
24	22	F	D244N	5.3	96	*ERT 5–10
25	42	F	G325D	2.6	29	ERT > 10
26	62	F	Gly325Asp	2.7	17	ERT > 10
27	40	F	A143T	0.3	141	CT < 5
28	59	F	R118C	0.1	101	naive
29	24	M	W277X	18	620	ERT 5–10
30	42	M	R227Q	15	92	ERT > 10
31	34	M	V296E	14	1107	*ERT > 10
32	26	M	G325D	17	177	* ERT > 10
33	20	M	G325D	5.3	8.7	ERT < 5
34	25	M	Y134D	6.9	6.4	ERT > 10
35	53	M	Y207C	12	413	* ERT > 10
36	65	F	N215S	0.6	24	ERT 5–10
37	40	F	c.718_719del	0.3	21	ERT >10
38	45	F	R363H	0.3	21	ERT 5–10
39	23	M	c.966DelC	-	-	ERT 5–10
40	43	M	c1033_1034delTC	77	1033	ERT < 5 y
41	34	M	A143T	0.9	34	ERT < 5
42	65	M	C223Y	20	231	ERT > 10
43	58	M	c.777del	17	33	* ERT > 10
44	58	M	c.717_718del Frameshift	19	1257	* ERT > 10
45	48	M	c.1072_1074del	12	38	ERT 5–10

**Table 2 cells-12-02102-t002:** Cardio characteristics in FD cohort. Variables are presented as mean ± STDEV or percentages ± STDEV. ^&^ A patient after heart transplantation was excluded from the study.

	No HCM		Borderline		HCM Mild		HCM Moderate/Severe	
	F (n = 7)	M (n = 4)	F (n = 3)	M (n = 3)	F (n = 11)	M (n = 7)	F (n = 3)	M (n = 7)
Age	30.7 ± 11	33.4 ± 14	54 ± 14	42.6 ± 15	43.6 ± 13	32 ± 12	50 ± 13	47 ± 14
Age min	18	18	39	27	22	21	40	24
Age max	46	53	68	58	63	54	65	65
LVPWd	0.7 ± 0.1	0.8 ± 0.05	0.8 ± 0.1	0.9 ± 0.06	1.08 ± 0.08	1.1 ± 0.05	1.3 ± 0.15	1.4 ± 0.1
LV mass	84.6 ± 16	116.6 ± 17	83 ± 4	120 ± 11	118.1 ± 29	173 ± 31	243	282 ± 169
LVEF %	62.3 ± 3	61.5 ± 2	56.3 ± 2	62.6 ± 5	59.3 ± 3	57.1 ± 5	54.0 ± 8	54 ± 3
EKG	normal	normal	abnormal	abnormal	abnormal	abnormal	abnormal	abnormal
Fibrosis	no	No	no	no	5	2	1 ^&^	5

Abbreviations: F—females; M—males; LVPWDd, cm—left ventricular posterior wall end diastole and end systole; LVmass, g—left ventricular mass; LV mass/BSA gm/m^2^—LV mass indexed to body surface area; LVEF—left ventricular ejection fraction; RVEF—right ventricular ejection fraction; LVIDd and LVIDs—left ventricular internal diameter end diastole and end systole; IVSd– interventricular septal end diastole; LVED—left ventricular end-diastolic pressure; LVEDd—left ventricular end-diastolic diameter; N/A—no data; Fibrosis—number of patients with fibrosis.

**Table 3 cells-12-02102-t003:** TGF-β1, active-TGF-β, VEGF-A, and FGF2 in healthy control vs. FD patients in gender-divided groups. Statistical evaluation includes mean ± STEV, *t*-test, and F-test scores. 95% confidential interval.

Biomarker	CNT vs. FD		CNT vs. FD Females		CNT vs. FD Males	
	FD	CNT	FD	CNT	FD	CNT
TGFβ1 95% CI	38.7 ± 4.1	12.5 ± 1.4	34.7 ± 4.7	13.9 ± 1.8	43.4 ± 7.1	10.9 ± 2.3
	^T test^ *p* < 0.0001 17.3 to 35		^T test^ *p* < 0.0001 10.4 to 31		^T test^ *p* < 0.0001 16.9 to 47.9	
Active TGFβ 95% CI	148 ± 11	111 ± 15	125 ± 16	81.93 ± 13	174 ± 16	145 ± 28
	^T test^ *p* < 0.05 −2.2 to 74.8		^T test^ *p* < 0.05 −1.1 to 87.2		^T test^ *p* = 0.1	
VEGF-A 95% CI	217.3 ± 54.6	71.4 ± 9.5	163.8 ± 62.4	64.7 ± 13.2	278.4 ± 92	79.6 ± 14
	^T test^ *p* < 0.005 26.1 to 265.7		^T test^ *p* = 0.11, ^F test^ *p* < 0.0001		^T test^ *p* < 0.05 −12 to 410	
FGF2 95% CI	90.7 ± 8.9	63.3 ± 6.8	95.2 ± 15	57.3 ± 6.2	85.8 ± 8.5	69.5 ± 12.9
	^T test^ *p* < 0.005 4.8 to 50.1		^T test^ *p* < 0.05 3.1 to 72.5		^F test^ *p* = 0.14	

**Table 4 cells-12-02102-t004:** Comparing inflammatory biomarkers, growth factors, and plasma/urine Lyso-Gb-3. * Pearson correlation, two tails. ** Pearson correlation, one tail.

	Plasma Lyso-Gb-3		Urine Lyso-Gb-3	
	Significant	Pearson Correlation	Significant	Pearson Correlation
TGFβ1	Yes	* *p* = 0.02, R = 0.3	Yes	* *p* = 0.007, R = 0.4
Active TGFβ	No		Yes	* *p* = 0.02, R = 0.3
VEGF-A	Yes	** *p* = 0.005, R = 0.3	No	
FGF2	No		No	

## Data Availability

The data presented in this study are available on request from the corresponding author.

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
