# Peer review of "Circulated TGF-β1 and VEGF-A as Biomarkers for Fabry Disease-Associated Cardiomyopathy"

_cells, 2023, doi:10.3390/cells12162102_

Round 1
Reviewer 1 Report
The topic of the article is interesting and presented comprehensively and exhaustively.
The statistics are appropriate, the graphs and tables easy to interpret
Author Response
We thank the reviewer for their careful reading of our manuscript and positive feedback.
Reviewer 2 Report
There are few patients with this disease (obviously), but being a case-control study, the healthy controls could have been increased to equal the sick.
Another aspect is that in the statistical program, together with the standard deviation, the confidence limits of the mean could have been calculated.
Author Response
There are few patients with this disease (obviously), but being a case-control study, the healthy controls could have been increased to equal the sick.
Response:
The control group consisted of age-and gender-matched healthy individuals without Fabry diseases, cardio, or kidney pathology. Moreover, blood was collected from healthy control only once, meaning the study was set up not as a pure “case-control study”.
Another aspect is that in the statistical program, together with the standard deviation, the confidence limits of the mean could have been calculated.
Response:
We included 95% confidence intervals in the table
Reviewer 3 Report
The article by Ivanova et.al., evaluates the potential circulating biomarkers, which can predict adverse cardiovascular events in Fabry disease-associated cardiomyopathy patients. The authors performed a set of experiments on serum samples of these patients to show that TGF-β, VEGF, and FGF2 are all increased in FD patients, which correlates and confirms the presence of persistent cardiac inflammation and fibrosis. However, these biomarker profiles differ significantly between FD men and women.
General comments (in no order of magnitude)
· In line 465, during the discussion, the authors describe that females have fibrosis progresses continuously, and delayed hypertrophy, hence TGF b (also active) is upregulated more in females. However, if you look at Table 2, only one female is shown to have fibrosis. Also, what does this symbol “&” indicates in Table 2?
· Often in HCM patients, PICP levels are found to correlate with fibrosis. I would suggest authors look at the level of PICP in the FD patient population. The combination of TGF-b and PICP may provide the highest prognostic benefit in the prediction of early cardiac fibrosis.
· (+) HCM FD has higher TGF-b in Figure 5 1A and Figure 5 1D. No need to repeat the result explanation in Line 286.
· In Figure 1A, can you plot the graph with different shapes for male and female data points. It will be easy for the readers. Do this for all the graphs whenever it is possible.
· Keep the color code consistent Figure 1E.
Author Response
The article by Ivanova et.al., evaluates the potential circulating biomarkers, which can predict adverse cardiovascular events in Fabry disease-associated cardiomyopathy patients. The authors performed a set of experiments on serum samples of these patients to show that TGF-β, VEGF, and FGF2 are all increased in FD patients, which correlates and confirms the presence of persistent cardiac inflammation and fibrosis. However, these biomarker profiles differ significantly between FD men and women.
General comments (in no order of magnitude)
- In line 465, during the discussion, the authors describe that females have fibrosis progresses continuously, and delayed hypertrophy, hence TGF b (also active) is upregulated more in females. However, if you look at Table 2, only one female is shown to have fibrosis. Also, what does this symbol “&” indicates in Table 2?
Response:
In the study, five female patients with a mild form of HCM and one female with a moderate form of HCM were diagnosed with cardio fibrosis. We and others postulate that it is very common that early signs of cardio fibrosis in Fabry females are missing due to delayed development of hypertrophy (compare with males), while fibrosis is already present in the female heart at a non hypertrophic disease stage.
In addition: we include the new reference in line 465 (Niemann et al., 2011).
Question:
Also, what does this symbol “&” indicates in Table 2?
Response:
We added an explanation in the text: “Simbol & is the individual patient who was excluded from the study after heart transplantation. A female patient was recruited to the study before heart transplantation, and blood was collected on her first visit; however, the patient was excluded from the study after heart transplantation.
- Often in HCM patients, PICP levels are found to correlate with fibrosis. I would suggest authors look at the level of PICP in the FD patient population. The combination of TGF-b and PICP may provide the highest prognostic benefit in the prediction of early cardiac fibrosis.
Response: It has been published that PICP elevated in Fabry disease and correlated with cardiomyopathy (Aguiar et al., 2018). Thank you for the suggestion; we will look PICP level in our cohort in the future.
- (+) HCM FD has higher TGF-b in Figure 5 1A and Figure 5 1D. No need to repeat the result explanation in Line 286.
Response: We removed the repeatable sentence (Line 286).
- In Figure 1A, can you plot the graph with different shapes for male and female data points. It will be easy for the readers. Do this for all the graphs whenever it is possible.
Response: When male/females were present with different colors and shapes in one graph, the graph became very busy and hard to read. Additionally, it was not clear how to present statistical data. To simplify, we included an extra graph separating females and males with different colors, shapes, and text.
- Keep the color code consistent Figure 1E.
Response: All graphs were created and synchronized in GraphPad using the “bright winter” color.
Aguiar, P., Azevedo, O., Pinto, R., Marino, J., Cardoso, C., Sousa, N., Cunha, D., Hughes, D., and Ducla Soares, J.L. (2018). Biomarkers of Myocardial Fibrosis: Revealing the Natural History of Fibrogenesis in Fabry Disease Cardiomyopathy. J Am Heart Assoc 7. 10.1161/JAHA.117.007124.
Niemann, M., Herrmann, S., Hu, K., Breunig, F., Strotmann, J., Beer, M., Machann, W., Voelker, W., Ertl, G., Wanner, C., and Weidemann, F. (2011). Differences in Fabry cardiomyopathy between female and male patients: consequences for diagnostic assessment. JACC Cardiovasc Imaging 4, 592-601. 10.1016/j.jcmg.2011.01.020.